# Orthodontic Status and Association with Oral-Health-Related Quality of Life—A Study of 16-Year-Old Norwegians with a Cleft Lip and Palate

**DOI:** 10.3390/ijerph21050550

**Published:** 2024-04-26

**Authors:** Paul K. Saele, Manal Mustafa, Anne N. Åstrøm

**Affiliations:** Oral Health Centre of Expertise/Western Norway and Department of Clinical Dentistry, University of Bergen, 5009 Bergen, Norway; manal.ibrahim.mustafa.sharafeldin@vlfk.no (M.M.); anne.aastrom@uib.no (A.N.Å.)

**Keywords:** OHRQoL, OHIP-14, cleft lip and palate, dental, orthodontics

## Abstract

Objective: To assess the association between clinical orthodontic indicators and oral-health-related quality of life, adjusted for socio-demographic factors, focusing on 16-year-old patients with cleft lip and/or palate (CL/P). Participants: One hundred and twenty-two patients with CL/P, representing cleft-lip (CL), cleft-palate (CP), unilateral/bilateral cleft-lip-palate (UCLP/BCLP), enrolled in the national CLP-Team, Bergen, Norway. Method: A cross-sectional study by two orthodontists assessing the number of teeth, intermaxillary sagittal relation (ANB-angle), dental arch and occlusion of 16-year-old patients with CL/P. All completed a digital questionnaire including self-reported socio-demographic variables, OHIP-14 questionnaire and dental aesthetics. Cross-tabulations with Pearson’s Chi-square test were used to identify associations between self-reported OHRQoL and socio-demographic and clinical indicators. Multiple variable analyses were conducted with binary logistic regression analysis using the odds ratio (OR) and 95% confidence interval (CI) to assess associations between OHRQoL and clinical indicators adjusted for socio-demographic variables. Ethical approval was granted by the regional ethics committee. Results: Patients with UCLP and BCLP had poorer clinical indicators compared to patients with CL and CP (*p* < 0.05). A total of 80% of the patients had OHIP-14 > 0. The highest oral impact was reported for psychological domains and articulation and the least for functional domains. Respondents with BCLP and those with poor intermaxillary relationships (ANB < 0°) reported a high impact on OHRQoL (*p* < 0.05). No statistically significant associations between other clinical indicators and socio-demographic variables such as gender, educational aspiration, and place of residence were reported. Conclusions: The study revealed an association between severe cleft diagnosis, missing teeth, misaligned teeth, negative overjet, and poor OHRQoL, but a statistically significant association was found only between OHRQoL and poor intermaxillary sagittal relations (unfavorable profile). To improve OHRQoL among patients with clefts, there is a need for an individual follow-up and prioritization of oral healthcare.

## 1. Introduction

Clefts of the lip and/or palate (CL/P) are among the most common congenital craniofacial anomalies [1,2] and are associated with a wide range of dental deviations [3,4]. A visible orofacial defect might impact the oral-health-related quality of life (OHRQoL) [5,6,7]. Few studies have explored the association between OHRQoL and clinical orthodontic indicators, such as missing teeth, misaligned teeth, and malocclusion, among individuals with CL/P.

Rehabilitation of a patient with CL/P tends to achieve a good facial appearance, speech, hearing, and dental aesthetic, and occlusion through a multidisciplinary approach from birth to adulthood [8,9,10]. In addition to anatomical deviations, patients with CL/P could have additional medical diagnoses that could cause challenges in offering them proper dental care [11]. In the typical patient with CL/P, the maxilla fails to develop normally, resulting in a retrognathic midface and a negative overjet, but with great individual variations [12,13,14]. Patients with clefts have relatively higher orthodontic treatment needs than non-afflicted people due to the higher incidence of agenesis and dental malocclusions [3,15,16,17]. The incidence of CL/P in Norway is 1.8/1000 live births [18,19]. All individuals born with a cleft are included in the national interdisciplinary team of medical specialists in Oslo, or in Bergen. Most of the treatment by the Bergen CLP team is undertaken in early adolescence, allowing evaluation of the treatment outcome at the age of 16 years [20].

Locker [21] introduced the concept of OHRQoL in 1988. An OHRQoL inventory, OHIP-49, was developed to measure oral impacts which could affect individuals’ daily activities [22]. A shorter version (OHIP-14) was later introduced [23]. This version has been translated into Norwegian and validated among the adult population [24,25]. OHIP-14 is a useful tool for evaluating the OHRQoL in different groups of patients [22,23,26], including individuals born with CL/P [27,28,29,30,31]. Kortelainen and co-workers [28] found a considerably poorer OHRQoL among Finnish children with CL/P than among their peers without clefts in OHIP-14 scores, especially those related to social well-being. To our knowledge, there are no reports of OHRQoL focusing on patients with CL/P in Norway.

The aim of this study is to assess the association between clinical orthodontic indicators and oral-health-related quality of life adjusted for socio-demographic factors, focusing on 16-year-old patients with a cleft lip and/or palate (CL/P).

## 2. Materials and Methods

This is a cross-sectional study focusing on patients with CL/P born between 1999 and 2003 who were treated by a national CLP team in Norway (Bergen, Norway). All patients had previously undergone clinical evaluation by two calibrated cleft orthodontists at an interdisciplinary consultation day at age 16. The calibration is performed annually and includes inter- and intra-rater reliability. After the exclusion of individuals with a syndrome, 205 with clefts were invited by letter or phone to participate. The study was voluntary, and each patient was given written information about the study. Consent was obtained from 153 patients. For statistical analyses, patients were stratified into four cleft types: unilateral cleft lip and palate (UCLP), bilateral cleft lip and palate (BCLP), isolated cleft of the palate (CP), and isolated cleft of the lip (CL).

A digital questionnaire with an estimated answering time of 10 min was distributed to all 153 patients. The questionnaire included socio-demographic information, in terms of gender, place of residence, and educational aspirations. Further, the questionnaire included one question about satisfaction with dental aesthetic and the Norwegian version of OHIP-14 to assess oral impacts on participants’ daily activities [24].

Place of residence was assessed as living in a rural (0) or urban (1) area. The educational aspirations were specified into ambitions of the number of years from no studies to more than five years of studies after high school. For statistical analysis, the response scale was dichotomized into <3 years (0), or ≥3 (1) years of studies.

The Norwegian version of OHIP-14 consists of 14 questions about problems with articulation, sense of taste, pain, eating, self-consciousness, feeling tense, dissatisfaction with diet, interruptions to meals, relaxing, feeling embarrassed, irritable, daily activity, less satisfaction, or inability to function, experienced by the subjects during the last year. For example, “During the last year, have you had trouble pronouncing any words because of problems with your teeth or mouth?” Each question was assessed using a Likert scale, graded as never (0), very seldom (1), sometimes (2), fairly often (3), or very often (4). The total sum score of the 14 questions varies from 0 to 56. Cronbach’s alpha for the original sum score was 0.89. The higher the sum score, the worse the oral-health-related quality of life. For statistical analyses, the Likert scale for each OHIP-14 question was dichotomized into no impacts, OHIP-14 = 0, or any impacts, OHIP-14 > 0 = 1. A sum score was constructed from the dichotomized items (0–14) and dichotomized into 0 (no impact) and 1 (at least one impact).

Satisfaction with dental aesthetics was assessed by the question “How satisfied are you with the aesthetics of your teeth?”. Responses were assessed by five items on a Likert scale ranging from very dissatisfied (0) to very satisfied (4). For analytical purposes, satisfaction with dental aesthetics was dichotomized into dissatisfied (scores of 0–1) = 0 and satisfied (scores of 2–4) = 1.

The data collected at the consulting day at age 16 include dental study models, photographs, panoramic x-rays, intraoral X-rays of the anterior segment, and a lateral cephalogram. The dental report is assessed by two experienced and calibrated orthodontists and includes cleft diagnosis, orthodontic treatment undertaken, missing teeth (agenesis or extracted due to poor quality), and orthodontic status based on the Bergen orthodontic grading index [20] (Appendix A). This grading index describes the dental and facial status and the possible need for later orthodontic or prosthodontic treatment, or the need for orthognathic surgery based on the index of orthodontic treatment need [32].

The Bergen orthodontic grading index presented in Appendix A includes: (1) sagittal intermaxillary relation (facial profile based on ANB-angle), (2) dental arch and measuring spacing or crowding in the dentition, (3) dental occlusion and, if present, negative or positive overjet. The measurements are ranked on a five-point scale from very severe deviation (Grade 1) to a perfect situation with no deviations (Grade 5). For analytical purposes, the sagittal intermaxillary relation (ANB-angle) is dichotomized into an unfavorable facial profile with an ANB < 0° = 0 (Grades 1, 2, and 3), or a favorable facial profile with an ANB ≥ 0° = 1 (Grades 4 and 5). The dental arch is dichotomized into the need for dental treatment due to spacing, crowding, or misaligned dentition = 0 (Grades 1, 2, and 3), or acceptable to perfect dentition and no need for dental treatment = 1 (Grades 4 and 5). Finally, the dental occlusion is dichotomized into the need for orthognathic surgery due to a negative overjet or open bite = 0 (Grades 1 and 2), or a positive overjet and good occlusion and no need for orthognathic surgery = 1 (Grades 3, 4, and 5).

### 2.1. Statistical Analysis

The data were analyzed using Statistical Package for Social Sciences version 24 (SPSS, Inc., Chicago, IL, USA). Frequencies, means, and standard deviations were used to describe socio-demographic characteristics and (selected) clinical orthodontic indicators of the participants. Cross-tabulations with Pearson’s Chi-square test were used for unadjusted bivariate analyses. Multiple variable analyses were conducted with binary logistic regression analysis using the odds ratio (OR) and 95% confidence interval (CI) to assess associations between OHRQoL and clinical indicators adjusted for socio-demographic variables. Socio-demographic factors and clinical indicators significantly associated with OHRQoL in the unadjusted analysis were included in the multiple logistic regression model, setting the level for statistical significance to *p* < 0.05. A two-way interaction term between clinical indicators and gender was tested for significance to explore any difference between boys and girls in the association of clinical indicators and OHIP-14.

### 2.2. Ethical and Legal Considerations

Ethical approval was granted by the regional ethics committee. All data were processed without a name, and a code number was used to link the questionnaire to the records from each participant at the 16-year consulting day. It is not possible to identify the participants from the results of the published study. The study was funded by Vestland County Oral Health Centre of Expertise.

## 3. Results

### Sample Profile

The socio-demographic variables of study participants according to cleft type is presented in Table 1. One hundred and twenty-two patients responded to the questionnaire (60% of those invited to participate). Boys and girls were equally represented (50.8% boys and 49.2% girls) and all types of clefts were represented in the sample: UCLP (26.2%), BCLP (9%), CL (27.9%), and CP (36.9%). Girls had higher frequencies of CP than boys (66.6% versus 33.3%), and boys had a higher frequency of CL than girls (52.9% versus 47.1%). A total of 64.8% of the participants were living in urban areas. Aspirations for education were high, and 67.2% of the total sample intended to undertake studies for three years or more. Aspirations for studies more than three years were the most frequent among patients with CL (85.3%) and the least frequent among patients with CP (64.4%).

Table 2 presents the frequency distribution of clinical indicators according to socio-demographic variables. There were no statistically significant differences in clinical indicators between boys and girls, rural or urban residences, or patients with aspirations for shorter or longer education.

Appendix A depicts the frequency distribution of clinical indicators according to all cleft types. More than half the participants (54.1%) had missing teeth (agenesis or extracted due to poor quality), 34.4% an ANB < 0°, 32% spacing and misaligned dentition, and 13.9% had a negative overjet or open bite. All clinical indicators were significantly poorer among the UCLP and BCLP groups than among the CL and CP groups (*p* < 0.05). Most of the study participants (95.1%) received orthodontic treatment prior to the 16-year clinical assessment, but none of the respondents had undergone orthognathic surgery or dental implant treatment at this age (data not shown in table).

Table 3 presents the impact scores of each item of the OHIP-14 inventory, stratified by cleft type and gender. The OHIP-14 items most- and least-frequently affected in the total study group were ‘articulation’ (49.2%) and ‘unable to function’ (7.4%), respectively. The OHIP-14 item most frequently affected was articulation in the UCLP (56.3%), BCLP (72.7%), and CP (55.6%) groups, but pain was reported among those with CL (61.8%). All cleft types reported minor impacts for ‘sense of taste’ and ‘unable to function’. The findings indicated minor differences between boys and girls, except that twice as many girls as boys reported being ‘unable to function in daily life’. The mean OHIP-14 score was 6.43 (SD = 7.65), the range was 0–36, and the median was 2.84. Patients with BCLP had the highest mean (10.45), while the other subgroups had mean scores ranging from 5.71 (CP) to 6.72 (UCLP) (data not shown in table).

Appendix A depicts the unadjusted association between satisfaction with dental aesthetics and OHRQoL. A total of 95% of participants who were dissatisfied with dental aesthetics versus 76.5% of those satisfied (0 = 0.042) reported any oral impact (OHIP-14 > 0), supporting the internal validity of the OHIP-14 index used in this study.

Table 4 presents the unadjusted/bivariate associations between OHIP-14 and cleft types, socio-demographic variables, and clinical indicators. The results reveal that 90.5% of patients with an ANB < 0° (poor intermaxillary relationship/unfavorable profile) versus 75% of those with an ANB ≥ 0° (acceptable intermaxillary relationship) reported any oral-health-related quality of life impact (OHIP-14 > 0) (*p* < 0.05). No statistically significant association was observed between the OHIP-14 score and cleft type, dental alignment, occlusion, missing teeth, or for the socio demographic variables in terms of gender, place of residence, or aspiration for education.

Table 5 depicts findings from multiple logistic regression analyses with the intermaxillary relationship (ANB-angle/facial profile) regressed on OHIP-14 and adjusted for socio-demographic variables in terms of gender, living in urban/rural areas, and educational aspirations. To adjust for possible confounding factors, socio-demographics were forced into the analysis, since no socio-demographic variable was statistically significantly associated with both OHIP-14 and clinical indices. In the final model, adolescents with an ANB < 0° (poor intermaxillary relationship) were 3.2 times more likely to report oral impacts (OR = 3.2, 95% CI 1.0–10.3) than those with an ANB ≥ 0° (acceptable intermaxillary relationship). No significant two-way interaction was observed between intermaxillary relations and socio-demographic variables upon OHIP-14, indicating that the associations between intermaxillary relations and OHIP-14 were similar across the socio-demographic groups investigated.

## 4. Discussion

This study revealed that patients with UCLP and BCLP had more missing teeth (agenesis or extracted teeth due to bad quality), a poorer intermaxillary sagittal relation (ANB < 0°/unfavorable profile), more misaligned dentition, and more negative overjet compared to patients with CL and CP. The majority of the study participants (80.3%) confirmed at least one oral impact or OHIP14 > 0. In the total sample and across the cleft types, the most- and least-frequently reported OHIP-14 items were problems with articulation and problems with function. Patients with missing teeth and a negative overjet/open bite were more likely that their counterparts to report any oral impact, but this association was not statistically significant.

A strength of this study is firstly the combination of clinical and self-reported measures of oral health in the same cohort treated after the same protocol in a national CLP team with a long tradition of cleft care and a high volume of patients [20]. In this respect, the present study constitutes a unique contribution to the research literature. Similar distributions of cleft diagnoses and gender have been observed in studies with larger sample sizes, indicating that the participants in our study are representative for this group of patients [33]. Secondly, a further strength of this study is the indication of acceptable psychometric properties in terms of the internal consistency reliability and internal validity of the OHIP-14 inventory. Thus, when comparing OHIP-14 with a global self-report measure of dental aesthetics, 95.8% of those being dissatisfied versus 76.5% of those being satisfied reported any impact according to OHIP-14. A limitation of the present study is nevertheless firstly a small sample size and secondly the lack of a comparable non-cleft control group for the OHIP-14 inventory in the Norwegian population at the same age. The present study is a single unit study with approximately 50–60 new cases per year. To achieve a cohort treated according to the same protocol and by the same orthodontists and within a reasonable timespan, the study group was limited to patients born between 1999 and 2003. This small sample size may have led to an underpowered statistical analysis. A third possible disadvantage is that our respondents (60% response rate) might not reflect the status of all the patients with CL/P who were initially invited to participate. The results might be biased since all respondents are treated at the same clinic that sent the questionnaire and felt compelled to please the examiner and this could potentially influence the results.

There are no studies of oral-health-related quality of life, using OHIP-14, in the general population of Norwegians between 16 and 20 years of age. In a sample of the Norwegian adult population aged 20–80 without clefts, Dahl et al. [24] reported a mean OHIP-14 score of 4.1 and a prevalence of OHIP-14 impacts of 65%. The present study shows higher mean OHIP-14 (6.4) and prevalence estimates, indicating a poorer OHRQoL in the Norwegian population with than without clefts. This is in accordance with similar studies in other cleft populations, reporting that overall OHRQoL is poorer among cleft patients than in the general population [28]. The highest prevalence of impact has previously been reported to be in the domains of psychological discomfort and psychological disability [27,28,34]. This is in accordance with findings in the present study, and in addition, those with a cleft involving the palate reported challenges with respect to articulation and speech. This corresponds with a previous study by Kummer [35] who described individuals with a cleft palate as being at risk of recurrent speech problems. From a Finnish cohort, Corcoran [27] reported no difference in the OHIP-14 score according to gender and the type of clefts. Furthermore, Foo and co-workers [36] in their study stated that treatments for orofacial clefts do not entirely remove the factors contributing to poor OHRQoL. In a study among 40-year-old patients with CL/P, Moi [6] reported that a common perception among patients was being different from their peers. However, having a cleft did not prevent them from achieving goals and satisfaction in life. In the present study, a high number of participants reported aspirations for longer education.

The degree of malocclusions and orthodontic treatment need have an influence on OHRQoL among patients without clefts [37]. For patients in need of orthognathic surgery, several studies have reported a poorer OHRQoL among cleft patients than in the population without clefts [38,39]. This is in parallel with the present findings confirming a poorer OHRQoL in the group with a negative overjet and high treatment need, compared to the respondents with good occlusion, though there were no statistically significant differences.

According to a review by Tsichlaki and co-workers [40], there is a lack of studies of large cleft populations and adequate control groups, especially studies comparing patients’ OHRQoL, demographic variables, and their orthodontic treatment needs. In the present study, the only statistically significant relation was the association between poor intermaxillary relation (ANB-angle < 0°/unfavorable facial profile) and poor OHRQoL. There was no statistically significant association between the severity of the cleft type (UCLP/BCLP versus CL/CP), number of missing teeth and degree of misaligned teeth (gaps in the dentition due to agenesis and the need for orthodontic and prosthodontic treatment versus no need for dental treatment), negative overjet (need for orthognathic surgery versus good occlusion), and poor OHRQoL in terms of high OHIP-14 scores.

## 5. Summary

This study revealed large individual variations regarding the clinical indicators and self-reported OHRQoL among 16-year-old Norwegian adolescents with a cleft lip and/or palate.

The results indicated an association between severe cleft diagnosis, missing teeth, misaligned teeth, negative overjet, and poor OHRQoL in terms of high OHIP-14 scores, but a statistically significant association was only observed between OHRQoL and poor intermaxillary sagittal relations (unfavorable profile). Enhancing oral-health-related quality of life (OHRQoL) in patients with clefts necessitates individualized follow-up and prioritization of dental treatment. Adequate resources must be allocated to ensure access to high-quality treatment for this patient group, which exhibits a greater dental treatment need compared to individuals without clefts.

## Figures and Tables

**Table 1 ijerph-21-00550-t001:** Socio-demographic distribution n (%) of study participants according to cleft types and in the total sample.

	UCLP n (%)	BCLP n (%)	CL n (%)	CP n (%)	Total n (%)
Total participants n (%)	32 (26.2)	11 (9.0)	34 (27.9)	45 (36.9)	122 (100)
Gender					
Boys	19 (59.4)	10 (90.9)	18 (52.9)	15 (33.3)	62 (50.8)
Girls	13 (40.6)	1 (9.1)	16 (47.1)	30 (66.6)	60 (49.2)
Residence					
Living in rural area	14 (43.8)	4 (36.4)	9 (26.5)	16 (35.6)	43 (35.2)
Living in urban area	18 (56.2)	7 (63.6)	25 (73.5)	29 (64.4)	79 (64.8)
Aspiration for education					
Aspiration for <3 years education	10 (31.2)	2 (18.2)	5 (14.7)	16 (35.6)	33 (27.0)
Aspiration for ≥3 education	22 (68.8)	9 (81.8)	29 (85.3)	29 (64.4)	89 (73.0)

**Table 2 ijerph-21-00550-t002:** Frequency distribution n (%) of participants’ clinical indicators according to socio-demographic characteristics.

	Missing Teeth (Agenesis or Extracted Due to Poor Quality)		Intermaxillary Relationship/Facial Profile		Dental Arch and Need for Orthodontic/Prosthodontic		Occlusion and Need for Orthognathic Surgery	
	Yes	No	ANB < 0°	ANB ≥ 0°	Spacing–Misaligned Teeth	Acceptable–Perfect Dentition	Negative Overjet or Open Bite	Positive Overjet Good Occlusion
Gender								
Male	33 (53.2)	29 (46.8)	24(38.7)	38 (61.3)	22 (35.5)	40(64.5)	11(17.7)	51(82.3)
Female	33 (55)	27 (45)	18(42.9)	42 (52.5)	17(28.3)	43(71.7)	6(10.0)	54(90.0)
Residence								
Rural	23(53.5)	20(46.5)	18(41.9)	25 (58.1)	15 (34.9)	28(65.1)	8(18.6)	35(81.4)
Urban	43(54.4)	36(45.6)	24(30.4)	55 (69.6)	24(30.4)	55(69.6)	9(11.4)	70(88.6)
Aspiration for education								
<3 years	21(63.6)	12(36.4)	13(39.4)	20 (60.6)	12(36.4)	21(63.6)	7(21.2)	26 (78.8)
≥3 years	45(50.6)	44(49.4)	29(32.6)	60(67.4)	27(30.3)	62(69.7)	10(11.2)	79 (88.8)

**Table 3 ijerph-21-00550-t003:** Frequency n (%) of each OHIP-14 item, total OHIP-14 > 0, and dissatisfaction with dental aesthetics, according to cleft types, gender, and in the total sample.

OHIP-14 Questions > 0	UCLP n (%)	BCLP n (%)	CL n (%)	CP n (%)	Boys n (%)	Girls n (%)	Total n (%)
Articulation	18 (56.3)	8 (72.7)	9 (26.5)	25 (55.6)	31 (51.7)	29 (48.3)	60 (49.2)
Sense of taste	5 (15.6)	2 (18.2)	1 (2.9)	4 (8.9)	6 (50.0)	6 (50.0)	12 (9.8)
Pain	10 (31.3)	7 (63.6)	21 (61.8)	19 (42.2)	30 (51.6)	27 (47.4)	57 (46.7)
Eating	4 (12.5)	4 (36.4)	6 (17.6)	12 (26.7)	12 (46.2)	14 (53.8)	26 (21.3)
Self-consciousness	17 (53.1)	7 (63.6)	13 (38.2)	14 (31.1)	30 (58.8)	21 (41.2)	51 (41.8)
Feeling tense	14 (43.8)	6 (54.5)	9 (26.5)	15 (33.3)	21 (47.7)	23 (52.3)	44 (36.1)
Diet unsatisfactory	5 (15.6)	3 (27.3)	5 (14.7)	9 (20.0)	9 (40.9)	13 (59.1)	22 (18.0)
Interrupted meals	5 (15.6)	3 (27.3)	4 (11.8)	5 (11.1)	10 (58.8)	7 (41.2)	17 (13.9)
Relaxed	8 (25.0)	4 (36.4)	7 (20.6)	8 (17.8)	12 (44.4)	15 (55.6)	27 (22.1)
Embarrassed	10 (31.3)	5 (45.5)	10 (29.4)	14 (31.1)	21 (53.8)	18 (46.2)	39 (32.0)
Irritable	6 (18.8)	3 (27.3)	5 (14.7)	7 (15.6)	10 (47.6)	11 (52.4)	21 (17.2)
Daily activities	4 (12.5)	3 (27.3)	4 (11.8)	5 (11.1)	7 (43.8)	9 (56.3)	16 (13.1)
Less satisfied	8 (25.0)	3 (27.3)	7 (20.6)	6 (13.3)	12 (50.0)	12 (50.0)	24 (19.7)
Unable to function	2 (6.3)	1 (9.1)	3 (8.8)	3 (6.7)	3 (33.3)	6 (66.7)	9 (7.4)
OHIP-14 total > 0	26 (81.3)	9 (81.8)	27 (79.4)	36 (80.0)	48 (77.4)	50 (83.3)	98 (80.3)
Dissatisfaction with dental aesthetics	6 (18.8)	3 (27.3)	9 (26.5)	6 (13.3)	13 (21.0)	11 (18.3)	24 (19.7)

**Table 4 ijerph-21-00550-t004:** Frequency distribution of participants n (%) OHIP-14 according to cleft types, socio-demographic variables, and clinical indicators.

	OHIP-14	
	No Impact n (%)	Impact n (%)
Cleft types		
UCLP	6 (18.7)	26 (81.3)
BCLP	2 (18.2)	9 (81.8)
CL	7 (20.6)	27 (79.4)
CP	9 (20.0)	36 (80.0)
Gender		
Male	14 (22.6)	48 (77.4)
Female	10 (16.7)	50 (83.3)
Residence		
Rural	6 (14.0)	37 (86.0)
Urban	18 (22.8)	61 (77.2)
Aspiration for education		
<3 years	7 (21.2)	26 (78.8)
≥3 years	17 (19.1)	72 (80.9)
Missing teeth (agenesis or extracted due to poor quality)		
Yes	11 (16.7)	55 (83.3)
No	13 (23.2)	43(76.8)
Intermaxillary sagittal relationship (facial profile)		
ANB < 0°	4 (9.5)	38 (90.5) *
ANB ≥ 0°	20 (25.0)	60(75.0)
Dental arch		
Spacing and misaligned dentition	8 (20.5)	31 (79.5)
Acceptable–perfect dentition	16 (19.3)	67 (80.7)
Occlusion		
Negative overjet or open bite	1 (5.9)	16 (94.1)
Positive overjet and good occlusion	23 (21.9)	82 (78.1)
Total		
122 (100)	24 (19.7)	98 (80.3)

* *p* < 0.05.

**Table 5 ijerph-21-00550-t005:** OHIP-14 regressed on intermaxillary relationship (ANB-angle/facial profile) adjusted for socio-demographics in terms of gender, living area, and educational aspirations.

	OR	95% CI
Gender		
Male	1	
Female	1.4	0.6–4.3
Residence		
Rural	1	
Urban	0.6	0.2–1.5
Aspiration for education		
<3 years	1	
≥3 years	0.9	0.1–10.8
Intermaxillary sagittal relationship (facial profile)		
ANB ≥ 0°	1	
ANB < 0°	3.2	1.0–10.4

## Data Availability

The data sets generated and analyzed in the study are not publicly available as they contain information that could potentially identify participants.

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
