# Peer review of "Orthodontic Status and Association with Oral-Health-Related Quality of Life—A Study of 16-Year-Old Norwegians with a Cleft Lip and Palate"

_ijerph, 2024, doi:10.3390/ijerph21050550_

Round 1

Reviewer 1 Report

Comments and Suggestions for Authors

This is an interesting, clear, and well written manuscript that I would recommend for publication. It is important that what is considered 'state of the art' management of a condition such as CLP be evaluated in terms of impact on OHRQoL. The findings, whilst not unexpected, are necessary to inform practice.

I have suggestions relating to study design, that cannot be changed at this stage but that the authors may wish to consider for future work.

The first is the exclusion of "individuals with syndrome". It is no longer acceptable to exclude patients in relation to disability. These patients appear nowhere in the literature and it would have been a perfect chance to "see" this group. OHIP can be used as a proxy tool if cognitive capacity was the perceived issue. I feel this is a missed opportunity.

The second is the restricted concept of sociodemographic profile. Socio-economic status and ethnicity are major factors that are missing here and that might (or not) have had an important influence on CLP associated Qol.

Very minor editing issues are:

Table 1, spelling of residence

Problems with additional spaces between certain words

Author Response

This is an interesting, clear, and well written manuscript that I would recommend for publication. It is important that what is considered 'state of the art' management of a condition such as CLP be evaluated in terms of impact on OHRQoL. The findings, whilst not unexpected, are necessary to inform practice.

I have suggestions relating to study design, that cannot be changed at this stage but that the authors may wish to consider for future work.

-The first is the exclusion of "individuals with syndrome". It is no longer acceptable to exclude patients in relation to disability. These patients appear nowhere in the literature, and it would have been a perfect chance to "see" this group. OHIP can be used as a proxy tool if cognitive capacity was the perceived issue. I feel this is a missed opportunity.

Answer: We thank the reviewer for drawing our attention to this part and we believe it is important to investigate OHIP among patients with syndromes. However, we were not able to collect records from the patients with syndromes in this cohort. but it will be taken into consideration in our future research work.

-The second is the restricted concept of sociodemographic profile. Socio-economic status and ethnicity are major factors that are missing here and that might (or not) have had an important influence on CLP associated Qol.

Answer: We agree with the reviewer that sociodemographic, and ethnicity is important factors, but this cohort is homogeneous with same ethnic Norwegian background, similar age, similar education (high school) with a social well-fare system that focus on minimal social disparities. This point can be addressed in future research with larger cohorts.

Very minor editing issues are:

Table 1, spelling of residence.

Answer: This is now corrected in table 1, page 3 line 137.

Problems with additional spaces between certain words.

Answer: Now addressed throughout the text.

Reviewer 2 Report

Comments and Suggestions for Authors

Thank you for submitting this manuscript for review. Overall the manuscript is well written and will be of interest to the scientific community. I have a few comments.

Methods:

1. In the paragraphs starting with line 97-122 the surveys are described and the Likert scale used for questions is also described. However the scales are different 0-4 vs 1-5. Why was there not one scale that was used for these questions?

2. in table 1, the sample size for BCLP is small (11). Was there consideration for combining the UCLP and BCLP groups for greater power? Due to the limited size of the BCLP group, some statements made in the results section should be tempered. This is mentioned as a limitation in the discussion. 

3.  In line 78-81, It is mentioned that the orthodontists are calibrated. Can you add how this was done and what is being calibrated?  Is there inter and intra-rater reliability data? 

Results:

1. For the small sample size for BCLP, the result mentioned in line 181-182 comparing UCLP and BCLP groups vs. CL and CP may need to be tempered as the power may not be present to make such a comparison, especially with BCLP. Unless the analysis was done with a CLP combined group.    

Comments on the Quality of English Language

Good. minor editing

Author Response

Reviewer 2.

Thank you for submitting this manuscript for review. Overall, the manuscript is well written and will be of interest to the scientific community. I have a few comments.

Methods:

  1. In the paragraphs starting with line 97-122 the surveys are described, and the Likert scale used for questions is also described. However, the scales are different 0-4 vs 1-5. Why was there not one scale that was used for these questions?

Answer: We agree with the reviewer that we should have used similar scale, now it is corrected to 0-4 scale, page 3, lines 111-114. The statistical analysis was re-performed to detect the association between cleft types and gender, presented same results as obtained from previous analysis.

  1. in table 1, the sample size for BCLP is small (11). Was there consideration for combining the UCLP and BCLP groups for greater power? Due to the limited size of the BCLP group, some statements made in the results section should be tempered. This is mentioned as a limitation in the discussion. 

Answer: We appreciate the reviewer's comment. It is indeed recognized in cleft studies that the incidence rates of bilateral cleft lip and palate (BCLP) diagnosis are relatively lower. However, it's important to note that BCLP represents a more severe defect with a potentially greater negative impact on Oral Health-Related Quality of Life (OHRQoL) compared to unilateral cleft lip and palate (UCLP). Therefore, combining these groups may obscure the possible negative impact of BCLP on OHRQoL relative to UCLP.

  1. In line 78-81, It is mentioned that the orthodontists are calibrated. Can you add how this was done and what is being calibrated?  Is there inter and intra-rater reliability data? 

Answer: The Bergen Cleft Lip and Palate (CLP) centre has diligently maintained comprehensive registrations and standardized records of all children born with CLP since 1962. These records are collected by the CLP team in Bergen during interdisciplinary consultations conducted at ages 6 and 16 for patients across all cleft subgroups. Annual calibration exercises ensure consistency among the two national CLP teams in Norway (Oslo and in Bergen). Unpublished data but accepted for revision on the agreement (ICCs) between the four orthodontists in the reevaluations was almost perfect for SNA and SNB (0.81 and 0.89, respectively) (table 1), Intra-rater agreement for the reassessments for the GOSLON index in in the range of 0.9-1-0 (table 2).

Notably, no specific calibration was conducted for this selected cohort, as it aligns with the routine practice of collecting and registering data for CLP patients managed by teams across Norway.

It is added in the text the practice in the national teams in Norway by annually calibrations. Page 2 lines 81-82.

Table 1. Absolute inter-judge agreement (individual ICCs) calculated using two-way random-effects models for intermaxillary sagittal relation (facial profile) measured from lateral cephalogram in terms of SNA, SNB and ANB angulations.

Variable

N

ICC (95% CI)

SNA angle

64

0.81 (0.70-0.88)

SNB angle

64

0.89 (0.84-0.93)

ANB angle

64

0.71 (0.56-0.82)

Abbreviations: ICC, intraclass correlation coefficient.

Table 2. Intra-rater agreement for the reassessments for the GOSLON yardstick performed by four independent and blinded orthodontists.

Variable

N

Percent agreement

(95% CI)

Cohens’ κa

(95% CI)

Gwet’s AC2

(95% CI)

Goslon yardstick

Orthodontist 1

64

0.98 (0.97-0.99)

0.86 (0.80-0.93)

0.93 (0.90-0.96)

Orthodontist 2

64

0.99 (0.99-1.00)

0.97 (0.94-1.00)

0.98 (0.96-1.00)

Orthodontist 3

64

0.96 (0.95-0.98)

0.83 (0.75-0.90)

0.89 (0.84-0.93)

Orthodontist 4

64

0.98 (0.97-0.99)

0.90 (0.85-0.96)

0.93 (0.90-0.97)

aCohen’s ordinal weighted kappa. Abbreviations: AC, Agreement Coefficient; CI, Confidence Interval.

Results:

  1. For the small sample size for BCLP, the result mentioned in line 181-182 comparing UCLP and BCLP groups vs. CL and CP may need to be tempered as the power may not be present to make such a comparison, especially with BCLP. Unless the analysis was done with a CLP combined group.    

Answer: A total of 122 patients completed the questionnaire, accounting for 60% of those invited to participate. The gender distribution was balanced, with 50.8% boys and 49.2% girls, and all types of clefts were represented: unilateral cleft lip and palate (UCLP) at 26.2%, bilateral cleft lip and palate (BCLP) at 9%, cleft lip (CL) at 27.9%, and cleft palate (CP) at 36.9%. It's important to note that the small sample size of BCLP cases may limit the statistical power for comparisons. However, we retained the classification of cleft types to assess the impact of each on the oral health-related quality of life (OHRQoL).

Reviewer 3 Report

Comments and Suggestions for Authors

This is a well-written manuscript and of interest to the dental professional community.   The research aim was to assess the association between clinical orthodontics indicators and OHRQL adjusted for selected socio-demographic factors among 16-year-old patients with CL/P. 

A few minor comments: 

Introduction

Please explain why age 16 was selected to evaluate the treatment outcomes. 

Discussion

Overall, the discussion section can be shortened by eliminating repetitive statements of the results in this section.   

Paragraph 2 - Numbering the strengths of the study can make reading easier and more organized. 

Line 261 - "Another possible disadvantage is that our respondents (60% response rate) might be biased..." Please discuss which types of biases could have occurred. 

Summary

A stronger statement on the uniqueness of the present study results is needed, including more specific clinical and public health recommendations. 

Author Response

Reviewer 3.

This is a well-written manuscript and of interest to the dental professional community.  The research aim was to assess the association between clinical orthodontics indicators and OHRQL adjusted for selected socio-demographic factors among 16-year-old patients with CL/P. 

A few minor comments: 

Introduction

Please explain why age 16 was selected to evaluate the treatment outcomes. 

Answer: Age 16 was chosen as a key milestone in the Bergen protocol in 1987 and used since then. By this age, most patients have typically completed their orthodontic treatment, allowing for evaluation of further dental interventions such as prosthodontics, implants, or orthognathic surgery in adulthood. Additionally, reaching patients for follow-up at this stage is more feasible before they potentially relocate for university or other pursuits.

Discussion

Overall, the discussion section can be shortened by eliminating repetitive statements of the results in this section.   

Answer: According to the reviewer suggestion the text in the discussion part is adjusted to delete the repeated results statement in page 8, lines 239-242, page 9, lines 277-279 and page 9, lines 300-303.

Paragraph 2 - Numbering the strengths of the study can make reading easier and more organized. 

Answer: Number/ numbering words is added in the text, page 8, lines 242-263.

Line 261 - "Another possible disadvantage is that our respondents (60% response rate) might be biased..." Please discuss which types of biases could have occurred. 

Answer: A low response rate for a questionnaire could introduce non-response bias, where those who choose not to participate differ systematically from those who do, potentially skewing the results and making them less representative of the entire population. This could also lead to self-selection bias, as certain types of individuals may be more likely to respond, distorting the findings. Additionally, since the patients were treated at the same clinic that sent the questionnaire, there may be a bias towards pleasing the examiners among respondents.

The text page 8, lines 261-263 are adjusted.

Summary

A stronger statement on the uniqueness of the present study results is needed, including more specific clinical and public health recommendations. 

Answer: “Enhancing oral health-related quality of life (OHRQoL) in patients with clefts necessitates individualized follow-up and prioritization of dental treatment. Adequate resources must be allocated to ensure access to high-quality treatment for this patient group, which exhibits a greater dental treatment need compared to individuals without clefts.”

Now this statement was added in the manuscript page 9, lines 310-314.